# The Neglected Insulin: IGF-II, a Metabolic Regulator with Implications for Diabetes, Obesity, and Cancer

**DOI:** 10.3390/cells8101207

**Published:** 2019-10-06

**Authors:** Jeff M. P. Holly, Kalina Biernacka, Claire M. Perks

**Affiliations:** Department of Translational Health Science, Bristol Medical School, Faculty of Health Sciences, University of Bristol, Learning & Research Building, Southmead Hospital, Bristol BS10 5NB, UK; mdxkz@bristol.ac.uk (K.B.); claire.m.perks@bristol.ac.uk (C.M.P.)

**Keywords:** IGF-II, insulin, metabolism, diabetes, obesity, cancer

## Abstract

When originally discovered, one of the initial observations was that, when all of the insulin peptide was depleted from serum, the vast majority of the insulin activity remained and this was due to a single additional peptide, IGF-II. The IGF-II gene is adjacent to the insulin gene, which is a result of gene duplication, but has evolved to be considerably more complicated. It was one of the first genes recognised to be imprinted and expressed in a parent-of-origin specific manner. The gene codes for IGF-II mRNA, but, in addition, also codes for antisense RNA, long non-coding RNA, and several micro RNA. Recent evidence suggests that each of these have important independent roles in metabolic regulation. It has also become clear that an alternatively spliced form of the insulin receptor may be the principle IGF-II receptor. These recent discoveries have important implications for metabolic disorders and also for cancer, for which there is renewed acknowledgement of the importance of metabolic reprogramming.

## 1. Introduction

Insulin-like growth factor-II (IGF-II) is one of the most abundant growth factors and, by far, the most abundant peptide with insulin-activity in the body. Some 21 years ago, I wrote an editorial commenting that, despite this, remarkably little was known regarding the physiology of IGF-II [1]. Although, in the intervening period, considerably more information has been acquired in relation to IGF-II physiology. It still remains somewhat of an enigma. This review will attempt to address these issues by assimilating many of the new observations. The indications for IGF-II having an important role, particularly in metabolic disorders and cancer, will be discussed. Lastly, a hypothesis will be proposed suggesting a fundamental role for IGF-II in human physiology.

## 2. Impediments to Progress

Advances in our understanding follow extensive studies, but to undertake such studies, the first task is always to raise the required funding. This normally involves making a very solid case and convincing reviewers that there is a sufficiently interesting story to justify the allocation of funds for further study. This has been relatively straightforward for insulin and IGF-I but not for IGF-II. Insulin plays a central role in the very prevalent human disease, diabetes mellitus. This was recognised around a century ago and resulted in intense study, leading to many ‘firsts’ in endocrinology and a string of Nobel Prizes [2]. The pivotal role for IGF-I as a key regulator of somatic growth was recognised in the 1950s [3] and many subsequent studies identified large variations in circulating IGF-I concentrations that were related to growth, nutrition, and different pathologies. In contrast, circulating IGF-II levels vary very little throughout postnatal life or with pathology. The circulating concentration of IGF-II is generally unchanged even in non-islet cell tumor-induced hypoglycemia; a relatively rare clinical condition that is manifest by very clear metabolic effects arising from excess IGF-II is produced by aggressive tumors [4]. An additional issue is that the post-natal physiology of IGF-II in rats and mice is very different from that of humans [5], which makes it difficult to use evidence from these experimental models to justify studies in humans. A further impediment to progress in our understanding is the complexity of IGF-II biology. The IGF-II gene is much more complex than IGF-I and both are considerably more complex than insulin, which, in comparison, is relatively straightforward. The additional complexity of IGF-II extends to the binding proteins, three of which bind IGF-II with higher affinity than IGF-I, and to the cell receptors with which it interacts. Insulin, IGF-I, and IGF-II interact with each other’s receptors, but there is an additional receptor with specificity and very high affinity for IGF-II. The more complex biology, in combination with fewer studies, has meant that we know much less about IGF-II and it is far more difficult to build a compelling story to justify funding.

These factors help explain why IGF-II has been relatively under researched. A simple search on PubMed reveals that, for every paper published on IGF-II, there have been approximately three published on IGF-I and 30 published on insulin. This imbalance is virtually the inverse of the complexity of their genes and also the inverse of their relative abundance in the adult human body. It is apparent, however, that an incredibly intricate series of checks and balances have evolved to control the cellular activity of IGF-II; again far more complex than that for IGF-I and insulin. The evolution of additional complexity for IGF-II and several additional cellular controls, further than those for IGF-I and insulin, implies that the stringent control of IGF-II is extremely important for the cell and yet it has received the least interest from researchers of all the insulin-like family. Purely on teleological terms, it seems that our neglect of IGF-II may be ill judged.

## 3. IGF-II: An Imprinted Multifunctional Metabolic Genetic Loci

The human IGF-II gene is located on the short arm of chromosome 11 at 11p15.5 (Figure 1). The 67 amino acid IGF-II peptide shares around 47% amino acid homology with pro-insulin and the genes presumably originated from an early evolutionary gene duplication event. Immediately, 5′ of IGF-II is the insulin gene (INS), just 1.4 kilobase (kb) upstream. A relatively simple gene, with three exons coding for pre-pro-insulin, driven by one promoter. In contrast, the IGF-II gene spans 29.3 kb and consists of 10 exons, four of which code for pre-pro-IGF-II and can be driven from five different promoters. Adding to the impediments detailed above, the current knowledge regarding the IGF-II gene is not accurately or completely described in any of the publicly available genetic databases, which most geneticists use as roadmaps, as has been carefully detailed recently [6]. Immediately 3′ of the IGF-II gene, 128 kb downstream, is a non-coding gene H19. Linked to IGF-II by an imprinting control region (ICR) immediately before H19 that regulates the expression of both IGF-II and H19. Thus, the INS/IGF-II/H19 locus forms a functional unit. The intricacies of the different promoters, alternative splicing, and distinct classes of prepro-IGF-II proteins have been nicely reviewed recently [6]. Imprinted genes are those that are only expressed from one of the parental chromosomes. IGF-II was the first identified imprinted gene with expression in mice being restricted to the paternal allele [7,8]. This has been studied extensively in mice but much less in humans, even though it is clear that there are important differences between species. In rodents, IGF-II expression post-weaning is restricted to very few sites and circulating levels fall to very low levels. In contrast, in adult humans, transcription of IGF-II from the P1 promoter in the liver is not imprinted, but there is biallelic expression [9] and high circulating IGF-II levels are maintained throughout life. The imprinting of the IGF-II/H19 locus is controlled by the transcriptional regulator CCCTC-binding factor (CTCF). An 11-zinc-finger nuclear protein that binds to the ICR between IGF-II and H19, depending on the methylation of a differentially methylated region (DMR) within the ICR, and results in reciprocal imprinting of the two genes. Normally, the ICR is paternally methylated and maternally unmethylated permitting CTCF to bind to the maternal allele. The bound CTCF then forms dimers with CTCF that also bind to regions near the IGF-II promoters. This produces and stabilises intra-chromosomal loop structures and forms the scaffold for recruitment of repressors that prevent an interaction with enhancers downstream of H19 suppressing transcription from the maternal IGF-II gene [10,11,12,13]. In contrast, CTCF cannot bind to the methylated paternal allele, which provides no scaffold for the recruitment of repressors and a chromosomal loop brings the IGF-II promoter into close proximity to the downstream enhancers that drive expression of paternal IGF-II. The DMR in the ICR extends into the promoter for H19, which results in silencing of the paternal allele whereas, on the maternal allele, a loop enables the downstream enhancers to promote H19 expression. Genetic or epigenetic abnormalities in the ICR disrupt this reciprocal imprinting and can result in developmental disorders of growth [14]. Abnormalities leading to loss of methylation at the ICR cause downregulation of IGF-II expression and biallelic expression of H19 and this can result in Russell-Silver Syndrome, characterised by severe prenatal and postnatal growth retardation and an increased risk of subsequent metabolic syndrome [14]. In contrast, abnormalities leading to the gain of methylation at the ICR causes loss of imprinting (LOI) and over expression of IGF-II and down regulation of H19, which can result in Beckwith-Wiedeman Syndrome. This is an overgrowth disorder associated with neonatal hypoglycemia and an increased risk of childhood tumors [14,15]. Disruption of imprinting, however, does not necessarily result in pathologies as screens of normal healthy neonates have revealed that around 20% exhibited LOI of IGF-II [16,17]. There is not, however, a good correlation between imprinting status and expression of IGF-II, since the majority of neonates with LOI had normal levels of IGF-II expression [17]. This implies that, for a developmental abnormality to be manifest, either the imprinting disorder must occur at a critical developmental stage or that other factors are also necessary.

The product of the IGF-II gene is the precursor, prepro-IGF-II, which contains a 24-amino acid N-terminal signal peptide and a C-terminal E-domain, which can be glycosylated. Cleavage of the signal peptide results in pro-IGF-II and further proteolysis cleaves the mature IGF-II from the E-domain that is also secreted [18]. This post-translational processing is not absolute and a variety of precursors, often referred to as ‘big’ IGF-II, are secreted with around 13% of the IGF-II in the circulation present as pro-IGF-II and around 16% as pre-pro-IGF-II [19].

### 3.1. Preptin

In a search for other hormones secreted from pancreatic islet β-cells, a small 34-amino acid peptide was identified that was found to be derived from cleavage of the E-domain from pro-IGF-II and this was termed preptin. It was isolated from the secretory granules of β-cells and shown to be co-secreted with insulin [18].

### 3.2. H19

The H19 gene was the first imprinted, long non-coding RNA (lncRNA) to be identified [20]. lncRNA are regulatory RNAs longer than 200 nucleotides. H19 lacks a conserved open reading frame and is not translated but it is transcribed, spliced, polyadenylated, and widely expressed. Hence, it is presumed to function as an RNA [21]. Over the last few decades, a plethora of roles have been ascribed to H19 by way of modulation of expression of other genes either via H19 acting as a source for microRNA (miRNA) or as a sink, or a decoy, for sequestering miRNAs or via interactions with chromatin modifying proteins, RNA binding proteins, or other proteins. The H19 transcript of around 2.3 kb is transcribed from five exons with alternative splicing, which results in two alternative transcripts [22].

### 3.3. miR-675

Micro-RNA (miRNA) are generally short, 18–25 nucleotide long segments of RNA that regulate the translation of many genes by translational repression or mRNA degradation. The H19 transcript contains an miRNA-containing hairpin within its first exon that serves as a precursor for two different miRNA. These are generated by the actions of the RNAse enzymes, Drosha and Dicer, to yield miR-675-3p and miR-675-5p [23]. The excision of miR-675 from H19 is regulated by the RNA-binding protein human antigen R (HuR), which protects RNA from endonucleases such as Drosha and Dicer. It was shown that miR-675 could slow cell proliferation and restrict placental growth, potentially by targeting the IGF-I receptor (IGF-IR) [24]. This was consistent with a parental conflict hypothesis [25] and the concept that maternally expressed genes generally suppress embryonic growth. Subsequently, there have been many reports of different genes targeted by miR-675. The database Targetscan (www.targetscan.org) lists 2598 potential targets for miR-675-3p and 1352 for miR-675-5p and the database miRTarBase (www.mirtarbase.mbc.nctu.edu.tw) lists 56 gene targets for miR-675-3p that have been, so far, experimentally validated and 70 validated gene targets for miR-675-5p. Thus, while RNA that code for proteins generally produce a limited set of products with a defined function, the miRNA can potentially affect many processes by targeting the expression of many other genes and, presumably, the specificity of these seemingly promiscuous regulators will eventually prove to be context-specific.

The complete picture and relative importance of the different targets of miR-675 are a long way from being elucidated. However, within the context of a review of the function of the INS/IGF-II/H19 locus, there are a few discoveries that already merit a mention. The first came from the observation that miR-675 was expressed in the placenta exclusively at the gestational time when placental growth ceases and that this was associated with down-regulation of the IGF-IR [24]. In the rapidly developing embryonic tissues, miR-675 remained tightly repressed despite high expression of H19 and over-expression of miR-675 inhibited the growth of a variety of embryonic cell lines [24]. It was suggested that H19 acted as a large latent reservoir of miR-675 that could be mobilised to suppress growth in situations of stress. The excision of miR-675 from H19 was shown to be regulated by the stress-response RNA-binding protein HuR [24]. In response to stress, HuR is translocated from the nucleus to the cytoplasm, which could then expose H19 to processing by Drosha generating miR-675 to suppress IGF-IR. The functional interaction between miR-675 and IGF-IR also appears to be operational during the differentiation of mesenchymal stem cells [26]. In contrast to growth suppression via targeting the IGF-IR, miR-675 has been reported to enhance the proliferation of vascular smooth muscle cells via translational repression of phosphatase and tensin homology deleted on chromosome ten (PTEN) [27]. This is the phosphatase that dephosphorylates phosphatidylinositol-3,4,5-phosphate, which is a critical second messenger in the PI3K signaling pathway, known as one of the main signaling pathways activated by the insulin receptor (IR) and the IGF-IR [28]. In addition, miR-675 has been reported to promote the growth of colorectal cancer cells by targeting the tumor suppressor retinoblastoma (RB) [29]. A further reported target of miR-675 suggests a more general epigenetic role by targeting histone deacetylases (HDACs) 4, 5, and 6 [30] and this resulted in a feedback loop as the inhibition of the HDACs, which reduced the binding of CTCF to the H19 ICR resulting in reduced H19 expression [30].

### 3.4. H19 as a miRNA Decoy

In addition to being a source of two miRNA, along with several other lncRNA, H19 also sequesters many other miRNA acting as a competing endogenous RNA, which is a decoy or sponge for miRNA. One of the most well characterised interactions is with the first identified miRNA, the let-7 family. In humans, there are nine mature let-7 miRNA members encoded by 12 different genomic loci [31]. By acting as a molecular sponge, H19 modulates let-7 availability [32] and, in turn, let-7 binding destabilizes H19 resulting in a double-negative feedback loop [33,34]. In addition to being well characterised as tumor suppressors, let-7 plays an important role in insulin/IGF signaling and glucose metabolism; indeed the IR and IGF-IR are both targets of let-7 [35]. A similar feedback loop has been described for another miRNA, miR-141, that is sequestered by H19, with levels of H19 and miR-141 being negatively related and miR-141 suppressing H19 expression [36,37]. Acting as a sponge for miR-152 enables H19 to have a broader epigenetic effect, since a target of miR-152 is DNA methyltransferase 1 (DNMT1). Increased H19 can, via this mechanism, result in DNA hypermethylation [38]. With many miRNA sequestered by H19 and each miRNA having multiple potential target genes, this presents a large scope for H19 to participate in many complex regulation networks.

### 3.5. H19 Interactions with Proteins

In addition to the many functions mediated via binding to miRNA, there have been multiple reports of H19 affecting cell function by binding directly to several different proteins. In gastric cancer cells, H19 binds directly to p53 resulting in increased proliferation and reduced apoptosis [39]. This has important metabolic effects as p53 binds to the promoter region of FoxO1, which itself directly drives a number of gluconeogenic gene promoters and reduced H19 can, therefore, drive increased gluconeogenesis via p53 [40]. In addition to H19 binding directly to p53 and inhibiting its actions, p53 is a target for miR-675-5p, which is derived from processing H19. This results in decreased p53 levels [41]. This suggests a close and complex interaction between H19 and p53. In addition to the broader epigenetic effects of H19, acting as a miRNA sponge, it also has further potential as an epigenetic regulator via a number of direct protein interactions. A report of multiple binding sites on H19 for IGF-II mRNA-binding protein 1 (IMP1) suggests a post-transcriptional link between the IGF-II and H19 genes with H19 potentially involved in the nuclear export, cytoplasmic localisation, and translation of IGF-II mRNA and, hence, regulating IGF-II expression [42]. A further clue that H19 may function as a more general riboregulator came with the report that H19 binds to a K homology (KH)-type splicing regulatory protein (KSRP) [43]. This study suggested that H19 acted as a molecular scaffold that facilitated the association of KSRP with unstable mRNAs and promoted their decay in a PI3K/AKT dependent manner [43]. A number of cellular functions appear to be regulated by KSRP either due to promoting the decay of mRNA or the production of miRNA from their precursor RNA, including let-7 [44]. Yet, a further potentially important regulation has been implied by the report that H19 binds with an enhancer of zeste homolog 2 (EZH2) [45], which is a core component of the polycomb-repressive complex 2 (PRC2) that is recruited to specific chromatin regions where EZH2 acts as a histone H3 Lys 27 (H3K27) methyltransferase to repress specific gene expression. In this way, H19 inhibits E-cadherin expression in bladder cancer cells [45] and also inhibits distinct subgroup of the Ras family member 3 (DIRAS3) in cardiomyocytes [46].

The most well characterised epigenetic modification of DNA is methylation that is mediated by the action of three S-adenosylmethionine (SAM)-dependent DNA methyltransferases (DNMTs), of which DNMT1 is a target of an miRNA sponged by H19 as described above. The methyl donor in this reaction is SAM, which yields S-adenosylhomocysteine (SAH) as a by-product. The SAH is then a potent inhibitor of the DNMTs. This inhibition can only be relieved by the hydrolysis of SAH to homocysteine and adenosine and only one enzyme exists that can catalyse this reaction and relieve the inhibition of DNMTs and that is S-adenosylhomocysteine hydrolase (SAHH). As a critical regulator of DNA methylation, SAHH is involved in many developmental processes and has been associated with a number of inherited disorders [47]. This critical activity of SAHH is inhibited by H19, which binds directly to SAHH [37]. This suggests that H19 may play an important general role in epigenetic regulation. A further direct binding partner of H19 appears to be the methyl-CpG-binding domain protein 1 (MBD1), which binds to methylated DNA and recruits histone deacetylase (HDACs) and histone lysine methyltransferase-containing complexes that, by regulating repressive histone marks on differentially methylated regions, control chromatin compaction and gene silencing [48]. This enables H19 to have important effects on a number of imprinted genes, including repressing IGF-II expression [48].

There have been many other reports of H19 interacting proteins affecting a variety of cell functions [49]. These include the RNA binding protein HUR that has been implicated in the processing of H19 to miR-675, as described above [24].

### 3.6. 91H

The complexity of the system was extended in 2008 when it was identified that H19 was also transcribed in an antisense direction to produce a further non-coding RNA that was termed 91H [50]. The 91H transcript is expressed predominantly from the maternal allele, as with H19, and accumulates in breast cancer cells due to increased stability. Knock-down indicated that it was involved in maintaining IGF-II expression in trans on the paternal allele [50,51]. However, in human esophageal squamous cell carcinoma cells, knockdown of 91H suggested that it inhibited IGF-II expression and loss of 91H was associated with more aggressive cancers [52]. Other studies have found 91H expression to be associated with more aggressive cancer phenotypes and, in colorectal cancers, this was attributed to an interaction between 91H and heterogeneous nuclear ribonucleoprotein K (HNRNK), which is an RNA-binding protein [52]. Considering that 91H has been studied far less than H19, it would be surprising if there were not much more to be discovered regarding the potential RNA and protein binding partners and functions of 91H.

A smaller antisense transcript derived from the H19 locus was also identified and termed H19 opposite tumor suppressor (HOTS) [53]. The identified transcript was found to be expressed more widely than H19 and is also maternally imprinted, like H19. The transcript associates with polysomes and encodes for a 150-amino acid peptide. Antibodies raised to this peptide indicated a 17 kDa monomer, a 34 kDa dimer, and a 29 kDa isoform. The polypeptide localises to the nucleus and interacts with an enhancer of rudimentary homolog (ERH). Over-expression of HOTS in cancer cell lines inhibited their growth and HOTS expression was absent in Wilms’ tumors that showed loss of heterozygosity or LOI at IGF-II/H19 and, thus, appeared to function as a tumor suppressor [53].

### 3.7. miR-483

In 2005, a further intriguing transcript from the loci was identified with the cloning of miR-483 from human liver [54]. The miRNA is encoded within the IGF-II gene in an intron between exons 7 and 8 of the coding region. A pri-mir-483 encodes two miRNA at either end of the miRNA hairpin and, on opposite strands, generates miR-483-3p and miR-483-5p. It was shown that miR-483 is co-expressed with IGF-II and a positive feedback mechanism was shown to operate with miR-483-5p enhancing IGF-II expression [55]. During fetal development, IGF-II transcription is driven mainly from the paternal allele from promoters P2, P3, and P4. However, in the adult liver, IGF-II expression is biallelic-driven from promoter P1 (9). The expression of IGF-II, driven by miR-483-5p, is mediated by miR-483p binding directly with the 5′ UTR of P3 and promoting its interaction with the RNA helicase A (DHX9) [56]. In addition, miR-483 functions as a conventional miRNA and the miRTarBase website lists 181 gene targets that have, so far, been validated for miR-483-3p and 143 gene targets validated for miR-483-5p.

### 3.8. Other Transcripts

#### 3.8.1. The INS/IGF-II Overlapping Region (INSIGF) Read-Through

A further complexity was revealed when a novel transcript was identified, which was composed of a transcript originating from the insulin promoter and consisted of exons 1 and 2 from the insulin gene spliced together with exons 2 to 4 of the IGF-II gene. This read-through transcript was termed INSIGF and a longer transcript was also found that included exons 1 and 2 of the INS gene and exons 2, 3, 7, 8, and 9 of the IGF-II gene. Both of these transcripts were reported to be imprinted in a similar manner to IGF-II [9]. The shorter transcript appears to be translated into a protein containing the pre-pro-insulin signal peptide, the 30 amino acids of the INS B-chain, eight amino acids of the INS C-petide in addition to 138 amino acids coded in the IGF-II gene, and its expression was reported to be limited to the human fetal pancreas and the eye [9]. Further evidence for the expression of this protein came with a report that INSIGF was primarily expressed in beta cells in human pancreatic islets [57]; even though there have still been very few studies of the expression or function of these transcripts and potential peptides in humans. This transcript has, however, also been reported to encode an lncRNA that was differentially expressed in non-small-cell lung cancer (NSCLC) when compared to adjacent normal lung tissue and which positively regulated IGF-II expression in NSCLC cells [58].

#### 3.8.2. IGF-II Antisense

In an examination of the transcripts expressed in Wilms’ tumors, it was discovered that an additional transcript was also derived from the IGF-II gene due to transcription from the paternal allele on the P1 promoter in the opposite direction, which generates an IGF-II antisense-RNA (IGF-IIas) [59]. The IGF-IIas was over-expressed in Wilms’ tumors when compared to neighboring normal kidney tissues and was also found in a number of other childhood tumors [59]. Unlike a similar IGF-IIas identified in mice, the human IGF-IIas has an open reading frame that could potentially encode a putative peptide consisting of 273 amino acids [59]. A further implication that IGF-IIas may encode a protein came from the report that the IGF-IIas transcript was found predominantly in the cytoplasm and associated with polysomes [60]. However, to date, no protein has yet been identified. There have, however, been a number of reports that IGF-IIas may function epigenetically as a lncRNA with effects on cell proliferation and invasion [61], apoptosis [62], and angiogenesis [63]. All of these reports found that these effects of IGF-IIas were reversed by increased expression of IGF-II and were consistent with an effect of IGF-IIas to suppress IGF-II expression even though no actual mechanism for this has yet been confirmed.

### 3.9. IMPs

In addition to miRNA that post-transcriptionally regulate mRNA, generally by silencing or targeting for mRNA degradation, it is now clear that there are many proteins that interact with mRNA, which can either destabilise mRNA, cooperate with miRNA to silence mRNA, or they can have opposing effects by reducing miRNA binding or by stabilising mRNA. A family of three RNA binding proteins that interact with IGF-II mRNA have been identified and are termed insulin-like mRNA-binding proteins (IMP1-3) that share around 56% amino acid homology with each other [64]. These were originally described as zip-coding proteins, thought to control the cellular localisation of mRNA, but are now recognised to regulate not only the localisation but also the stability and translation of mRNA. It has been shown that IMP2 is phosphorylated by the mTOR complex 1 promoting its association with the 5′ UTR of IGF-II and enhancing its translation and then also promoting the ribosomal entry of IGF-II mRNA leading to increased IGF-II protein synthesis [65]. Contrary to the implication from their nomenclature, the IMPs do not only bind to IGF-II mRNA but also interact with numerous other different mRNAs affecting the translation of many genes [64].

### 3.10. GRP94

Molecular chaperones ensure the correct assembly and folding of proteins and the targeting of misfolded proteins for degradation. These include a number of endoplasmic reticulum (ER) stress proteins, including heat shock proteins and glucose-regulated proteins [66]. These proteins also maintain the integrity of the ER and the mitochondria. Most of the molecular chaperones are fairly promiscuous and ensure the folding of many client proteins. However, a glucose-regulated protein, GRP94, was identified as a chaperone for IGF-II [67,68] and only a few other secreted and membrane proteins. It was shown that GRP94 interacts with pro-IGF-II intermediates in the cell and is essential for the processing and secretion of IGF-II [67,68]. The critical role of IGF-II in skeletal muscle differentiation is also dependent on the activity of GRP94 [68,69]. GRP-94 is also a chaperone for IGF-I and, recently, a hypomorphic variant of GRP94 was identified in children with short stature and IGF-I deficiency [70]. GRP94 is glucose-regulated. Its levels increase in response to falling levels of cellular glucose, which clearly implicates a metabolic function. However, the significance of this to IGF-II and its role as a metabolic regulator are yet to be clarified. However, it seems clear that GRP94 is an essential component of the IGF-II system.

### 3.11. Receptors

#### 3.11.1. Insulin and IGF-I Receptors

As with the homology between the IGFs and proinsulin, their receptors are, likewise, very similar and are closely related members of the class II receptor tyrosine kinase family that share both structural and functional homology [71]. The IGF-IR and insulin (IR) receptors are both translated and the proteins are then cleaved to yield an extracellular α-subunit and a transmembrane β-subunit that are disulphide-linked. These then dimerise to form heterotetrameric mature receptors. The IGF-IR binds IGF-I and IGF-II with high affinity and has very little affinity for insulin. The insulin receptor exists as two isoforms, due to alternative mRNA splicing, with the IR-A isoform containing 12 fewer amino acids in the extracellular C-terminal domain of the α-subunit due to splicing excluding exon 11, whereas the IR-B isoform has these additional amino acids due to inclusion of exon 11. These extracellular α-subunits form the ligand binding domain and, hence, the alternative splicing affects ligand specificity. The IR-B predominantly binds insulin and has a much lower affinity for IGF-I/-II. Hence, this is the classic insulin receptor. The loss of 12 amino acids from IR-A subtly reduces the specificity and results in a relative increase in affinity for IGF-II [72]. As a consequence, IGF-II binds IR-A with an affinity approaching that of insulin, and IR-A also binds pro-insulin with high affinity, in contrast to the very low affinity for these ligands with IR-B [19,72]. Although IR-A binds insulin and IGF-II with similar affinities, there is evidence that the binding of each of these ligands activates different signaling pathways, which results in the differential regulation of gene expression and diverse cellular responses [73,74,75]. Stimulation of IR-A by IGF-II has been reported to result in more prolonged activation of ERK1/2, compared to insulin stimulation [76], and this may contribute to the greater mitogenic and less metabolic effects reported for IGF-II activation of IR-A [76,77]. In pancreatic islets, insulin stimulates insulin gene expression via the IR-A, and not the IR-B, but stimulates glucokinase via the IR-B. These differential effects are due to IR-A and IR-B residing in different lipid raft domains within the plasma membrane [78,79]. The potential role of IR-A as an IGF-II receptor has important implications, especially when considering the large differences in the relative abundance of IGF-II and insulin in the body.

The α-/β- dimers of the insulin and IGF-IR are so similar that they hetero-dimerise to form hybrid receptors, both IR-A/IGF-IR and IR-B/IGF-IR hybrids. The relative abundance of these hybrids depends on the relative expression of each receptor in cells that express both IR and IGF-IR. As a consequence of the lack of discrimination in dimer formation, the less abundant receptor will be present primarily as a heterodimer rather than as a homodimer. These heterodimer hybrid receptors appear to predominantly act as IGF-I receptors [80,81], but there is still much to be learned regarding their physiology.

#### 3.11.2. IGF-II Receptor

There is also a very specific IGF-II receptor (IGF-IIR) that is a single large transmembrane protein and is structurally and functionally completely different from the other IGF receptors [82,83]. The paternal imprinting of IGF-II is counterbalanced by the maternal imprinting of the IGF-II receptor gene in the mouse [84], although this does not appear to have been conserved in humans in whom the expression of this gene does not appear to be imprinted [85]. The IGF-II receptor binds IGF-II with a very high affinity but is very specific and has very little affinity for IGF-I or insulin. This receptor is generally considered not to act as a traditional signaling receptor in response to IGF-II binding but acts as a clearance receptor for IGF-II, internalizing and directing IGF-II to lysosomes for degradation. Thus, this controls cell exposure to IGF-II. Consistent with this role was the observation that disruption of IGF-IIR gene expression in mice resulted in elevated circulating levels of IGF-II and overgrowth [86]. The cellular location of the IGF-IIR is dynamically regulated by insulin in the same manner as insulin regulates the glucose transporter, GLUT4 [87,88,89]. These receptors are mainly intracellular but are rapidly translocated to the cell surface in response to insulin stimulation. This dynamic relocation of IGF-IIR has been known for more than 30 years but still little is known regarding the physiology. The dynamic relocation in response to insulin could represent a novel means for metabolic control, like the dynamic translocation of GLUT4, even though such a metabolic role has yet to be actually demonstrated. When nutrient abundance triggers pancreatic insulin release, the insulin could result in internalization of the IGF-IIR, which leads to less IGF-II clearance and, hence, more IGF-II is available to interact with the IGF-IR and the IR-A, which then compliments the direct actions of insulin itself. Actual experimental evidence to support this potential role is yet to be obtained. The IGF-II receptors are, furthermore, clearly multifunctional and have several other functions in addition to being clearance receptors for IGF-II. Their most well-characterised role is as a mannose 6-phosphate receptor involved in the targeting of lysosomal enzymes to the lysosomes within the cell [82,90]. These receptors, however, also bind latent transforming growth factor-β (TGF-β) and enable its activation on the cell surface. They also bind to retinoids, urokinase-receptors, and many other proteins. Much has still to be learned regarding the functional consequences of all of these interactions with the IGF-II receptor and their relation to IGF-II physiology is still unknown [82,83,90].

### 3.12. IGF Binding Proteins

In humans, whereas insulin circulates in a free unbound state, the IGFs bind with high affinity to six binding proteins (IGFBP-1 to IGFBP-6). The IGFBPs do not bind to insulin and are unrelated to the cell-surface receptors but are structurally closely related to each other [91]. The six IGFBPs all have very distinct functional properties and they are produced in different quantities and combinations in different tissues [91]. The IGFBPs sequester the IGFs immediately that they are secreted from cells and considerably slow their clearance. This enables very high concentrations of IGFs to accumulate in the body. In the circulation, two of the IGFBPs, IGFBP-3 and IGFBP-5, are bound to a further large glycoprotein, which is known as the acid labile subunit (ALS) and which is present in excess. This ternary complex is too large to cross capillaries and, hence, is retained in the circulation and further slows clearance such that, in adult humans, the total IGF-I and IGF-II concentration in the circulation is around 100 nanomolar. This is around 1000 times higher concentration than that of insulin and, while insulin levels fluctuate acutely in response to metabolic conditions, the circulating concentrations of IGFs are very stable due to the very long half-life of these complexes [91]. In the tissues, IGF concentrations are less than 20% of that in the circulation [92], which is still a large excess over that needed for cell regulation.

All of the insulin in the body is secreted from pancreatic islets and, although the IGFs are expressed in most tissues, the majority of the IGFs present in the circulation originate from the liver where the production of IGFs and IGFBP-3 are very dependent on the nutritional status [93]. The high, stable levels of circulating IGFs, therefore, provide a large pool of metabolic regulators that reflect chronic metabolic status. At the cellular level, optimal activation of the insulin and IGF receptors are both achieved with just one to two nanomolar concentrations, which indicates that there is a vast excess of IGFs in the circulation. Hence, while the activity of insulin throughout the body is largely determined by the rate of secretion from the pancreas, the constitutive secretion of the IGFs within any tissue is just one of the determinants of the total amount of IGF that the cells are exposed and the control of IGF-activity is much more complex [5,91]. The IGFs bind to the IGFBPs with affinities higher than that of the IGF-IR and IR-A receptors, so most of the IGF in the body has restricted availability for receptor activation. There is considerable evidence that IGFBPs can not only sequester IGFs away from cell receptors and restrict activity, but that they can also promote activity at the cellular level via a variety of mechanisms [91] and also enhance delivery of IGFs to specific tissue compartments [94]. Activity in a tissue is, therefore, not necessarily determined by the secretion rate of IGFs and not necessarily determined by total IGF concentration [5]. In the circulation, IGFBP-1 levels undergo a marked circadian variation due to dynamic insulin regulation of its production in the liver. This appears to provide additional acute control to ensure that IGF-activity is modulated in a manner that is appropriate to prevailing metabolic conditions [95].

## 4. Indications for a Role in Diabetes and Obesity

Clearly defined roles have been established for insulin and IGF-I, helped by the large body of literature documenting big variations in their circulating levels with age, nutrition, and different physiological and pathological conditions. In contrast, studies of IGF-II levels have found very little variation in any condition [1]. There have, however, been some indications that serum levels of IGF-II are related to nutritional status, since levels have been reported to be increased in obesity [96]. The raised IGF-II levels in obesity also appear to be reversible with weight loss [97]. There are many other different strands of evidence indicating potential roles for IGF-II in human physiology and, most consistently, these suggest a role for IGF-II as a metabolic regulator.

The systemic consequences of excess production of IGFs due to specific tumors imply that there are clear distinctions in the pathophysiology of IGF-I and IGF-II. There are no common tumors that produce sufficient IGF-I to increase circulating levels, but systemic levels are increased in acromegaly due to pituitary tumors of the GH-producing cells that result in stimulation of excess hepatic production of IGF-I. The symptoms of this condition reflect the growth promoting actions of IGF-I; particularly gigantism if presenting in childhood. In contrast, there is a clinical syndrome associated with over-production of IGF-II directly from tumors called non-islet cell tumor-induced hypoglycemia. The systemic symptoms of this condition are not associated with tissue over-growth but with disturbed metabolism, principally severe episodic hypoglycemia [4]. With its insulin-like activity, IGF-II acts on the liver to reduce hepatic glucose output and increase glucose storage as glycogen. In muscles, IGF-II also helps decrease the blood glucose level by facilitating glucose uptake and oxidation and by stimulating the synthesis of both lipids and proteins [98].

Unlike observational studies, reports of associations of pathologies with genetic variants are unlikely to indicate confounding or reverse causality and many different genetic studies have consistently implied a causal role for IGF-II in diabetes. Several cohort studies have found associations between inter-individual genetic variations in the INS/IGF-II/H19 locus and body weight and obesity [99,100,101,102] as well as with abdominal and visceral fat [103]. In addition, the methylation status of the IGF-II DMR is associated with body weight and adiposity [104,105]. An intriguing recent report examined genetic variants in all protein-coding regions that were associated with type 2 diabetes in people of Latino descent, which is a population that has a very high rate of this disease. They identified a variant within the IGF-II gene that disrupted the intron 4 to exon 5 splice acceptor site, which had an allele frequency of 17% in the Mexican population, but which was very rare in European populations [106]. Heterogeneous Latino carriers of the variant had a 22% decreased risk of type 2 diabetes and homogeneous carriers had a 40% reduced risk. Lower expression of the variant was associated with lower HbA1c in patients with type 2 diabetes [106]. This is particularly intriguing since the alternative splicing would only be predicted to affect transcripts derived from promoter 2. However, current data indicates that such transcripts are minimally expressed in human tissues [6].

That these genetic variants have functional consequences is supported by reports that circulating IGF-II concentrations are also associated with weight, the waist-hip ratio, and weight-gain [107,108,109]. Furthermore, high in utero expression of the paternal IGF-II allele may be related to fat deposition postnatally in the offspring [107], and the level of IGF-II methylation at birth may contribute to the development of obesity and weight gain in early childhood [107]. The decline in circulating IGF-II levels in mid-life is also associated with adiposity in early old age [110], which, again, implies that IGF-II has a role in subsequent fat disposition.

A clinical case with a chromosomal breakpoint upstream of the IGF-II gene, separating the gene from some of its telomeric enhancers, has been reported to result in intra-uterine growth retardation (consistent with the recognised role of IGF-II in fetal development) but also resulted in the development of atypical early type 2 diabetes that was associated with insulin resistance and a marked increase in abdominal adiposity [111]. This case would also be consistent with an important role for IGF-II in metabolic regulation and especially adiposity and insulin resistance.

### 4.1. Preptin

Serum levels of preptin (derived from pro-IGF-II) are increased in obese individuals [112] and in subjects with type 2 diabetes [113]. These studies would be consistent with the co-secretion of preptin with insulin from pancreatic β-cells and the ability of preptin to enhance insulin secretion [18,114]. This may contribute to the hyperinsulinemia associated with type 2 diabetes.

### 4.2. miR-483

In two recent studies of subjects with type 2 diabetes, circulating levels of miR-483-5p were found to correlate with fasting insulin levels, HbA1c, and a measure of insulin-sensitivity [115] as well as with fasting insulin levels and body mass index (BMI) [116].

### 4.3. H19

Genetic variance in the H19 gene is related to the risk of type 2 diabetes with bioinformatics suggesting that the relevant polymorphisms would affect H19-miRNA interactions [117].

### 4.4. IMPs

Genome-wide association studies have also consistently found associations between the risk of type 2 diabetes and a polymorphism in the IMP2 gene [118,119,120]. This polymorphism has also been associated with fasting insulin levels and measures of impaired beta-cell function [121]. Consistent with a role for IMP2 in the regulation of adiposity, silencing of IMP2 expression resulted in slightly smaller mice but these mice were then relatively resistant to obesity when fed a high-fat diet (126). These links between IMP2, type 2 diabetes, and obesity may not be mediated via altered translation of IGF-II since, like most RNA-binding proteins, the IMPs bind to many other different mRNA.

### 4.5. IGFBPs

Serum levels of IGFBP-2 have consistently been associated with measures of fat mass, central adiposity, and insulin-resistance [108] as well as with the metabolic syndrome and cardiovascular risk factors [122].

### 4.6. Receptors

Abnormal circulating levels of an extracellular domain shed from the IGF-II receptor have been observed in patients with obesity and type 2 diabetes [123].

Changes in insulin receptor isoform expression are associated with metabolic pathologies, with IR-A expressed to a higher degree than IR-B in liver [124] of diabetic monkeys. This would imply that these metabolic tissues become more IGF-II responsiveness in perturbed metabolic conditions. Early reports of insulin receptor isoforms in humans were inconsistent likely due to methodological issues. With improved tools/technology, more recent studies have been consistent and all indicate that metabolic tissues become more IGF-II responsive with the development of insulin resistance and metabolic dysregulation. In a study of IR isoforms in subcutaneous adipose tissue, a high BMI was associated with an increase in the IR-A/IR-B ratio and the ratio correlated with fasting insulin levels [125]. Furthermore, weight loss induced by a low-calorie diet or bariatric surgery resulted in a relative increase in IR-B [125]. Similarly, a study found that the IR-A/IR-B ratio was increased in liver samples from obese subjects with type 2 diabetes compared to those with normal glucose tolerance and, following bariatric surgery, the ratio normalized due to a reduction in the IR-A isoform in the individuals whose diabetes was resolved [126]. In addition, an impaired insulin response in patients with myotonic dystrophy patients is associated with a lower level of IR-B in muscle tissue [127,128].

## 5. Metabolic Effects of Components of the IGF-II Locus

There are many reported metabolic effects of the different components produced from the IGF-II locus (Figure 2) that could help explain how the locus is involved in diabetes and obesity, as described above. Many actions appear to be due to a coordinated regulation between mRNA, lncRNA, microRNA, and proteins. The integration of these will need considerably more work to completely unravel.

### 5.1. IGF-II

In terms of abundance within the body, IGF-II is, by far, the most prevalent insulin-like peptide present throughout an entire human life. This is not the case in rodents. One of the routes through which IGF-II was independently discovered was due to its ‘insulin-like’ activity. Using an adipose tissue assay measuring stimulation of glucose uptake and very specific insulin-antibodies to deplete all of the insulin present in serum, they found that 93% of the insulin-like activity was not due to insulin but due to another peptide that they termed non-suppressible insulin-like activity (NSILA) [129]. This was, subsequently, renamed IGF-II when structural characterization revealed considerable homology to pro-insulin [130]. For many years, it was thought that IGF-I and IGF-II activate the same cell surface IGF-I receptor but the clear distinctions in their physiology raised puzzling questions. The distinction between the two IGFs is most clear in early development. IGF-II plays a very important role in fetal and early neonatal growth and development. This role appears to be conserved in humans very similarly to that defined in rodent models. In utero, IGF-II has an important role in the control of nutrient partitioning in the placenta and fetus [131]. In rodents, there is a clear switch at weaning when the expression of IGF-II throughout the body virtually ceases and there is a clear end to its major systemic developmental role. In humans, and other higher mammals, this developmental switch does not occur, and IGF-II remains the most prevalent IGF throughout life. The evidence outlined above implies that the maintained high levels of IGF-II play an important metabolic role. The evidence that IGF-II is an effective activator of IR-A provides the mechanism whereby IGF-II could act on metabolic tissues that is distinct from IGF-I. A consideration, of the relative abundance of INS/IGF-I/IGF-II and the tissue distribution of the different receptors, implies that, during normal fasting conditions, IGF-II would be the predominant activator of IR-A and only in the post-prandial state would insulin activate both IR-A and IR-B [19].

In addition to its expected insulin-like activity, IGF-II could have important metabolic actions via its role in the development and maintenance of important metabolic tissues. It has been recognized for a long time that the INS/IGF-system is essential for the development of adipocytes [132]. The proliferation of preadipocytes is stimulated by IGF-I [133,134] and the presence of insulin-like stimulation is essential for the differentiation of preadipocytes into mature adipocytes [135]. In mature human adipocytes, IGF-II has the expected insulin-like actions and stimulates glucose uptake. However, IGF-II stimulates this with lower potency than insulin, but this has to be interpreted in the context of the relative abundance of these peptides in the body [136]. In humans, there is a marked functional distinction between adipocytes in different anatomical regions with visceral adipocytes being much more closely associated with metabolic syndrome, type 2 diabetes, and cardiovascular risk [137]. There is accumulating evidence that IGF-II has a specific role in this differential role of adipose tissue depots. Although visceral adipocytes are considered to be less insulin responsive, they express higher levels of the insulin receptor (IR), but this increase is entirely due to the IR-A isoform [132]. In addition, primary cultures of visceral adipocytes, when compared to subcutaneous sections, have increased GLUT4 levels and greater insulin-stimulated glucose uptake [138,139]. A depot-specific effect of IGF-II has also been reported with IGF-II reducing the differentiation of preadipocytes from visceral adipose tissue but stimulating that of subcutaneous adipocytes [140]. This was accompanied by an IGF-II induced decrease in GLUT4 and in IR-A in the visceral adipocytes [140].

The other major insulin-responsive metabolic tissue is skeletal muscle and there is even more evidence that IGF-II plays an essential role in the development and maintenance of skeletal muscle mass. The differentiation of stem cells into post-mitotic myotubes is potently stimulated by IGF-II [141]. The key myogenic transcription factor stimulates autocrine production of IGF-II [142] and the IGF-II then works with MyoD in an amplification cascade to promote muscle differentiation [143]. In mature skeletal muscle, IGF-II again has the same insulin-like effect and stimulates glucose uptake [144].

The important role that IGF-II plays in placental function [131] may also have implications for metabolic disorders. Although there appears to be little transfer of insulin across the placenta, the transfer of IGF-II/H19 locus products has yet to be determined [145]. The effects of such products on the placenta, or placental transfer, may be relevant to the fetal macrosomia associated with diabetes in the mother or gestational diabetes. A recent report indicates that miR-483-3p contributes to macrosomia via effects on trophoblasts [146]. Altered fetal growth following assisted reproductive technology (ART) has also been attributed to the high maternal estradiol, induced by ART, which increases placental IGF-II via an epigenetic effect [147].

### 5.2. Preptin

Preptin was originally isolated from the secretory granules of pancreatic islet β-cells and is co-secreted with insulin [18,114]. Preptin also acts on pancreatic β-cells to enhance the glucose-mediated insulin secretion [18,114], but the relation of this activity to IGF-II physiology is yet to be determined.

### 5.3. miR-483

In contrast to the effect of IGF-II, which promotes the formation and maintenance of adipose tissue and skeletal muscle, the miRNA embedded within the IGF-II gene, miR-483, inhibits the proliferation and differentiation of bovine skeletal myoblasts [148] and murine adipocytes [149]. Furthermore, miR-483 levels in adult rats and humans are programmed by early life nutritional exposures [149]. In contrast, miR-483 promotes the differentiation of human adipocytes and miR-483 expression in adipose tissue is raised in subjects with multiple symmetric lipomatosis [150]. This is also consistent with the report that miR-483 levels are correlated with BMI [116].

### 5.4. H19

In addition to the data showing an association with genetic variance implying that H19 plays a role in diabetes [117], a screen of RNA expression in livers from diabetic mice revealed that H19 was the most altered lncRNA [151,152]. The expression of H19 correlated with that of gluconeogenic enzymes and silencing H19 resulted in an increase in their expression implying that H19 was a key regulator of hepatic gluconeogenesis in diabetes [152]. Further work from this group indicated that H19 regulated the transcription of FoxO1, which is an important transcription factor involved in regulating gluconeogenic enzymes [40]. Other work indicated that H19 contributed to the gluconeogenesis observed in diabetes by altering the methylation of hepatocyte nuclear factor 4a (HNF4a), which is another critical transcription factor that regulates key enzymes involved in gluconeogenesis [153]. In addition to its role in hepatic gluconeogenesis, there have also been reports that H19 plays an important role in the alterations of skeletal muscle insulin sensitivity observed in type 2 diabetes, potentially by targeting the key cell energy regulator 5′ adenosine monophosphate-activated protein kinase (AMPK) [154].

An important metabolic role for H19 could also be mediated by its function as an miRNA decoy. As such, H19 appears to be a critical regulator of let-7, that plays a pivotal role in regulating glucose metabolism and metabolic programming. As mentioned earlier, both the IR and the IGF-IR are targets for let-7 repression [33,35,155]. Metformin is the most commonly used drug used to treat type 2 diabetes. Metformin upregulates let-7 which then leads to repression of H19 and activation of SAHH and, as a consequence, widespread changes in DNA-methylation [34].

The H19/let-7 double-negative feedback loop appears to play an important role in regulating skeletal muscle development and insulin sensitivity. The differentiation of skeletal muscle is enhanced by depletion of H19, which increases the availability of let-7 [32]. In humans, the abundance of H19 was significantly decreased in the skeletal muscle of subjects with type 2 diabetes. This increased the availability of let-7 [33]. In rodent models, acute hyperinsulinemia downregulated H19 and this was due to increased production of let-7 resulting in H19 destabilisation [33]. In a high-fat diet model of obesity in mice, insulin resistance in the muscle was associated with a decrease in H19, an increase in let-7, and a decrease in two let-7 targets: the IR and lipoprotein lipase. Furthermore, H19 depletion in muscle cells within a culture resulted in impaired insulin sensitivity [33].

### 5.5. miR-675

There have been reports that miR-675, derived from H19, has effects on the formation of both adipose tissue and skeletal muscle, like miR-483. In murine models, miR-675 promoted skeletal muscle differentiation and regeneration [156]. In contrast, miR-675 levels in skeletal muscle were associated with a loss of muscle mass in patients with chronic obstructive pulmonary disease [157]. In vascular smooth muscle, miR-675 was associated with the proliferation and PTEN was shown to be a target of miR-675 [27]. Since PTEN is a key negative regulator of the PI3K/Akt pathway, which mediates the metabolic actions of insulin/IGFs; then, if miR-675 targets PTEN in other tissues, it could play an important role in metabolic regulation. In cardiomyocytes, miR-675 was found to target peroxisome proliferator-activated receptor-α (PPARα) [158]. Since PPARα is a ligand-activated transcription factor that regulates carbohydrate, lipid, and amino acid metabolism [159], then miR-675 could play an additional important metabolic role if it regulates PPARα in other tissues. The differentiation of mesenchymal stem cells into adipocytes is inhibited by miR-675 with histone deacetylases (HDAC) 4-6 being identified as targets for miR-675 [30]. In contrast, miR-675 was found to be consistently increased in a screen of miRNA that were altered during adipocyte differentiation from mesenchymal stem cells [160].

### 5.6. IMPs

In addition to affecting IGF-II expression, the links between IMP2 and diabetes could be due to many other potential effects of IMP2. One clearly relevant action is that IMP2 protects mRNA from let-7-dependent silencing of gene targets by binding to the mRNA at the corresponding let-7-binding site and preventing let-7 mediated target mRNA degradation [64]. As described above, let-7 targets several genes that play important roles in insulin/IGF signaling and glucose metabolism.

### 5.7. Receptors

The most important recent developments regarding receptors, which are relevant to the metabolic activity of IGF-II, have been related to the insulin receptor isoforms. Although IR-A has generally been reported to confer more of a mitogenic response, compared to the metabolic response elicited by activation of IR-B, the most important implication is that the expression of IR-A enables cells to be more IGF-II responsive, especially when considering the relative abundance of IGF-II compared to insulin within the body [19]. There have also been some other interesting observations regarding specific metabolic actions of IR-A. Immortalized neonatal hepatocytes with silenced insulin receptors were transfected to express either IGF-IR, IR-A, or IR-B and glucose uptake was examined. Silencing IR reduced glucose uptake and this was restored by either the IGF-IR or IR-A, but not by IR-B. This was shown to be due to IGF-IR and IR-A being able to associate with GLUT1 or GLUT2 and act as co-transporters to enhance basal glucose uptake [151,161]. Over-expression of IR-A specifically in the liver was also shown to be more effective than IR-B in ameliorating glucose intolerance in a mouse model of type 2 diabetes [162]. The significance of these findings to the role of IR-A and IGF-II in humans in relation to other tissues and in other stages of development are all yet to be investigated.

## 6. Role of IGF-II in Pancreatic Islet Function

The β-cells of the pancreas are the sole site of expression of insulin and play an essential role in maintaining glucose homeostasis in humans and β-cell failure leads to diabetes. Autoimmune destruction of β-cells leads to type 1 diabetes, whereas failure of the β-cells to compensate with an increase in insulin secretion in the face of rising glucose levels leads to type 2 diabetes. Maintaining β-cell mass is, therefore, central to metabolic control and there is considerable evidence that the IGF-II locus plays an important role in this. In a study, performing deep RNA sequencing of purified β-cells from 11 individuals, the most highly expressed transcript was INS but the second and third most highly expressed were the INSIGF read-through transcript and IGF-II. These represent 38%, 10%, and 2% of the β-cell transcriptome, respectively [163]. In early development and in response to stress, IGF-II plays an important role as a β-cell survival factor [164]. The proliferation and maintenance of β-cells are also regulated by the gluco-incretin hormones known as glucose-dependent insulinotropic polypeptide (GIP) and glucagon-like peptide 1 (GLP-1). Selective silencing of β-cell expression of IGF-II or immuno-neutralising IGF-II secreted from β-cells indicated that the effects of the gluco-incretin hormones were mediated via an autocrine IGF-II loop [165]. In these experimental paradigms, the glucose-stimulated insulin secretion from the β-cells was also markedly suppressed, which implies that this IGF-II autocrine loop plays an important role in pancreatic insulin secretion [165]. It has been reported that insulin promotes insulin transcription via activation of IR-A and promotes β-glucokinase transcription through activation of IR-B [78]. This would imply that insulin transcription would be particularly IGF-II responsive. In addition, the presence of IR-A on β-cells enhances their proliferative response to IGF-I and enhanced glucose uptake, potentially via a direct interaction between IR-A and the glucose transporters GLUT1 and GLUT2. Yet, the effects of IGF-II were not examined in this model [151]. In early type 2 diabetes, the pancreas compensates to maintain glucose homeostasis as insulin-resistance develops. This is associated with a large increase in β-cell mass and studies in mice, with β-cell-specific knock-out of IGF-II indicating that IGF-II contributes around 30% to this β-cell expansion [166]. In contrast, over-expression of IGF-II specifically in β-cells resulted in β-cell de-differentiation and endoplasmic reticulum stress as well as increased the susceptibility of mice to diabetes [167]. This indicates that too much IGF-II expression in β-cells may predispose them to the onset of diabetes [167].

Using rodent models, it has been reported that H19 has a role in regulating β-cell mass, but not insulin secretion, via its ability to sequester let-7, which results in a de-repression of let-7 target genes and activation of the PI3K/Akt pathway [168]. Pancreatic β-cell function may also be affected by preptin, which enhances insulin secretion [18].

A study that profiled 553 miRNA, to identify those that were differentially expressed between pancreatic islet insulin-secreting β-cells and glucagon-secreting α-cells, found that one of the most differentially expressed miRNA in β-cells was miR-483, derived from the IGF-II gene [169]. Over-expression of miR-483 in β-cells resulted in increased insulin transcription and secretion and protected β-cells from cytokine-induced apoptosis. Whereas, over-expression of miR-483 in α-cells decreased the transcription and secretion of glucagon [169]. In addition, the expression of miR-483 was found to be raised in islets taken from prediabetic db/db mice (a model of obesity, diabetes, and dyslipidemia) with expanded β-cell mass implying that miR-483 may play an important role in pancreatic compensation during the development of diabetes [169].

## 7. IGF-II Locus and Cancer

The IGF-II locus also plays a very important role in the development of numerous cancers and, in many ways, this mirrors the critical role that IGF-II/H19 plays in early embryonic development. The concept that cancer cells may be embryonic rests was proposed back in the 19^th^ century [170]. More recently, this concept has resurfaced in a modified form with new developments in our understanding of stem cells and cell differentiation plasticity. It has become apparent that there are patterns of cell behavior that are programmed within all cells, but many of these are normally only expressed during embryogenesis or wound healing. These same processes can, however, be inappropriately reactivated in neoplastic cells, either in response to cell stress or as the cell reverts back in terms of a differentiation status [171,172,173]. A critical remaining question is whether the changes observed in IGF-II/H19 are a consequence of the genetic/epigenetic alterations that occur in neoplastic cells or whether epigenetic alteration in IGF-II/H19 predisposes tissues to the development of cancer. The common occurrence of LOI in Wilms’ tumors and other childhood malignancies implies that, at least in these cases, the epigenetic alteration in IGF-II leads to a predisposition for these cancers [7,174]. The other issue that suggests that the IGF-II/H19 might be fundamental to many cancers relates to the ancient metabolic role that this locus appears to serve. The importance of metabolism to the development of cancers was a central theme of cancer research in the early 20th century and was re-discovered and belatedly added to the ‘hallmarks of cancer’ earlier in this century [175].

There have been many recent reviews of the role of IGF-II in cancer [174,176,177] and, therefore, we will only make limited comments and also discuss other products of the IGF-II locus.

### 7.1. IGF-II

LOI of IGF-II was first identified in Wilms’ tumors [178] and childhood tumors associated with the over-growth condition Beckwith-Weidemann syndrome [179]. LOI has, subsequently, been observed in many different cancers, including Ewing sarcoma, rhabdomyosacrcoma, clear cell sarcoma, renal cell sarcoma, hepatoblastoma, glioma, testicular, colorectal, gastric, esophageal, laryngeal, pancreatic, bladder, breast, prostate, testicular germ cell, and gynecological cancers with varying frequencies in different series, but often as high as 40% to 70% [180]. In addition, there have been reports of loss of heterozygosity for IGF-II, H19, CTCF, and the IGF-IIR [181], which indicates that many epigenetic alterations to the locus are common in cancers. It should, however, be born in mind that, as mentioned earlier, LOI of IGF-II has been found in around 20% of normal healthy neonates [16,17]. The loss of the IGF-II imprint has also been reported to extend into adjacent normal tissue and not simply be confined to tumor tissue, at least for colorectal [182,183], laryngeal [181], and prostate cancers [184]. The LOI of IGF-II in normal prostate tissue was found to increase with age in mouse and humans and this was more extensive in men with prostate cancer and, in mice, the age-related effect was associated with a decrease in CTCF levels and its binding to the IGF-II/H19 ICR [185]. These findings could either imply that a defect in the tumor is transmitted in a ‘field effect’ to the surrounding normal tissue or, more likely, that an epigenetic defect in the tissue predisposes it to the development of a cancer. This would be consistent with the changing concepts emerging from the explosion of new genetic data. It is now clear that mutations accumulate in normal epithelial cells with age and that potentially neoplastic cells are prevalent throughout such continually renewing epithelial tissues. However, although mutations are necessary for neoplastic transformation, they are not sufficient for the development of a cancer [186]. In addition to the neoplastic cell, or ‘seed,’ the local internal milieu, or ‘soil’, has to be fertile for a cancer to develop and an increase in IGF-II could provide the metabolic/mitogenic/survival stimulus that enables this. This concept is supported by the finding that, in a mouse model with increased expression of IGF-II, created by crossing mice with an Apc mutation (that predisposes to colorectal cancer) with mice in which the ICR upstream of H19 had been deleted (resulting in biallelic expression of IGF-II). These mice developed twice as many tumors and also manifest less differentiated normal colonic epithelium [187]. In a similar model, using mice with mutations in the CTCF-binding site at the IGF-II/H19 ICR, that prevented CTCF binding, biallelic IGF-II expression was observed and there was an increase in the prevalence and severity of prostatic intra-epithelial neoplasia [188]. In human screening colonoscopy studies, IGF-II LOI was associated with an increase in expression of IGF-II, a family history of colorectal cancer, and an increased risk of development of a colorectal adenoma [189,190]. These would be consistent with IGF-II LOI predisposing tissues to neoplastic development.

In contrast to the many epigenetic alterations affecting IGF-II commonly found in cancers, actual genetic mutations in IGF-II appear to be relatively rare with 149 mutations in 24 different cancers listed in the Cancer Genome Atlas (https://portal.gdc.cancer.gov/), with a prevalence ranging from 7% in uterine endometrial carcinomas to less than 0.5% in most of the common cancers [6]. The significance of these mutations in relation to IGF-II physiology or to cancer biology has yet to be determined.

The other emerging theme regarding IGF-II and cancer relates to its developmental role. When normal tissue stem cells undergo malignant transformation to form cancer stem cells, they acquire more differentiation plasticity, such as in fetal life when the IGF-II/H19 locus is a key regulator. As these cells differentiate these regulators may then activate progression through epithelial to mesenchymal transition (EMT) promoting tumor invasion and metastasis. The stem-cell-origin-of-cancer hypothesis postulates that mutations accumulate throughout life in normal tissues in the self-renewing stem cells and not in the mature differentiated somatic cells. Hence, these are the cells from which cancers originate. The malignant phenotype of increased growth, survival, and invasion can be explained as reactivation of inherent developmental programs within cells that are then hijacked to support tumor development. The reactivation could be via epigenetic means that reactivate the IGF-II/H19 locus. The evidence that IGF-II supports and promotes cancer stem cells has been extensively reviewed [19,180,191,192]. This evidence suggests that oncogenic transformation activates an autocrine feedback loop mediated via IGF-II, potentially via an epigenetic mechanism, and that this activates pluripotency factors such as Oct-4, SOX2, and Nanog and works with Wnt/β-catenin, Notch, and Sonic hedgehog pathways to maintain and regulate cancer stem cells. This evidence also indicates that activation of the IGF-II locus may promote EMT [191] that could then facilitate tumor invasion and metastasis. This is consistent with reports that IGF-II is associated with progression and prognosis [193,194,195,196]. A further implication of the reactivated developmental program is related to the splice variant of the insulin receptor, IR-A, which was originally reported to be highly expressed in fetal and cancer cells [77]. There is now considerable evidence that IR-A is highly expressed in many different cancers [19]. This would render the cancer cells more responsive to IGF-II and this appears to support increased proliferation, differentiation, plasticity, and metabolism [19]. The metabolic effects of IGF-II via IR-A may support the increased energy demands of the tumor, potentially via the ability of IR-A to act as a co-transporter with GLUT1 or GLUT2 to enhance basal glucose uptake [151,161].

### 7.2. Other Components of the IGF-II/H19 Locus

As described above, the IGF-II/H19 locus appears to be an evolutionary conserved and integrated functional unit with a fundamental role in development and metabolic control. If oncogenic transformation involves a reactivation of an inherent developmental program, then it would be expected that the entire IGF-II/H19 functional unit would be involved in the development of cancer. The emerging evidence suggests that this is the case. For each of the components, there have been increasing reports of their involvement in many different cancers (Figure 3) [29,38,45,49,51,52,197,198,199,200,201,202,203,204,205,206,207,208,209,210,211,212,213,214,215], but how all of the components of the genetic locus are integrated together in the development of cancers will require considerably more investigation to unravel.

There have yet to be many studies of miR-483, embedded within the IGF-II gene, even though reports are beginning to appear [203] with miR-483 being identified as an epigenetic modulator of IGF-II imprinting within tumors [216]. There have also been a few reports suggesting that the IGF-II antisense transcript may act as a tumor suppressor for cancers including prostate [61]. In contrast, there have been many studies regarding H19 and cancer. This includes its altered expression and potential role in cancer initiation, EMT, progression, and metastasis. These reports have been extensively reviewed recently [217,218,219]. There have been fewer investigations to date regarding the role of the antisense transcript 91H in cancers. Yet, the shorter translated transcript that produces HOTS was identified and reported to act as a tumor suppressor [53]. In relation to miR675, derived from H19, there have been reports of both positive [209] and negative [200] effects on tumor growth, progression, and invasion. There has also been a recent interesting report that miR675 can promote malignant transformation of mesenchymal stem cells by blocking DNA mismatch repair [220].

The IMPs, IGF-II-mRNA binding proteins, have been heavily implicated in several cancers and again may provide a link between normal development and cancer stem cell maintenance as previously reviewed [64]. It is still not clear, however, whether the activity of IMPs is completely dependent on IGF-II physiology since they have many other interesting actions, including a recent report that they destabilize progesterone receptors in breast cancer cells [221]. Similarly, the IGF-II molecular chaperone, GRP94, has been implicated in the proliferation, survival, invasion, and metastasis of cancers, as reviewed previously [222,223]. Yet, again, the extent that these roles involve IGF-II is yet to be clarified.

## 8. Hypothesis: A Fundamental Metabolic Role for IGF-II

Throughout most of their evolution, mammals were grazing eaters and their activity was dependent on continued consumption of energy supplies. In lower mammals, such as rodents, IGF-II played an essential role in early development when the fetus or neonate had a 24-h energy supply from the placenta or from the mammary glands. After weaning, when food ingestion was established, but in a grazing pattern, IGF-II is no longer generally expressed, and the pancreas becomes established as the primary metabolic regulator, with nutrients absorbed in the gut stimulating pancreatic insulin secretion. Higher mammals, including humans, have adopted intermittent feeding patterns, with protracted long periods between meals. This eventually allows humans to use these prolonged time periods constructively. In the post-prandial state, the pancreas is still the dominant metabolic regulator and, in grazing-eaters, they are mainly in the post-prandial state continuously. However, for most of their evolution, humans were only in the post-prandial state for very short periods. During the prolonged periods between meals, when there is little stimulation of the pancreatic islets, there may still be times of physical activity and other activity needing redistribution of nutrients and there may still be requirements for insulin-like regulation. We suggest that, with its much more complex regulation, the IGF-II/H19 locus has evolved to fill this role and act as a more general metabolic regulator with the capacity for considerable fine-tuning at the tissue level depending on context. When intermittent feeding patterns evolved, there was also the development of a specialized function for visceral adipose tissue due to its anatomical location, with venous drainage via the portal system, which could provide a supply of energy to the liver when required during the long periods between meals. As described above, the evidence suggests that IGF-II may act on visceral adipocytes, via the IR-A, to enable this specialized function, distinct from that of subcutaneous adipose tissue. This could help explain why depot-specific differences in adipocytes are much more pronounced in humans (in whom very high IGF-II levels are maintained throughout adulthood) when compared to rodents (in which IGF-II levels are minimal after weaning) [224].

The distribution of body fat has greater significance for the development of obesity-related morbidities than the simple extent of fat accumulation. Although the vast majority of fat is held in subcutaneous adipose tissue, visceral adiposity is more closely associated with the development of metabolic syndrome, type II diabetes mellitus, and cardiovascular disease [137]. Liposuction, which involves the removal of a large proportion of subcutaneous fat, has little effect on insulin sensitivity [225], whereas removal of visceral fat by omentectomy significantly improves insulin sensitivity independent of changes in total body weight [226]. The visceral adipocytes have a specialized function. They are more metabolically active and rapidly release nutrients during conditions of stress, which directly provide free fatty acids as substrates for hepatic glucose production and lipoprotein metabolism. This evolved specialist function for the IGF-II/H19 locus, to provide alternative metabolic fuel during prolonged periods between meals and at times of stress, may, however, be maladaptive during times of prolonged positive energy balance, which, throughout most of the evolution, would have been extremely rare. A western lifestyle with regular snacks and energy-dense foods, however, results in humans being in a postprandial state for 16–18 h per day [227] and, hence, a prolonged positive energy balance.

With neoplastic transformation, the reactivation of inherent developmental programs could lead to the cancer cells hijacking the developmental role of IGF-II/H19, which then operates with the inherent metabolic role to promote cancer progression. As in fetal development, the tumor has continuous metabolic demands and reactivation of the IGF-II/H19 locus could help satisfy these requirements, as it does in early life. Most of the components of the locus have been implicated in various cancers, but how IGF-II, preptin, HOTS, H19, 91H, miR-483, miR-675, IMPs, and the other components are integrated both during early development and in carcinogenesis will require considerably more work to elucidate. With the microRNA having dozens of potential gene targets and the lncRNA having even more potential interactions with proteins and many microRNA, each with their own multiple gene targets, an integrated picture of how the whole locus operates as a functional unit will require new approaches for understanding such an integration of multiple related regulators.

## Figures and Tables

**Figure 1 cells-08-01207-f001:**
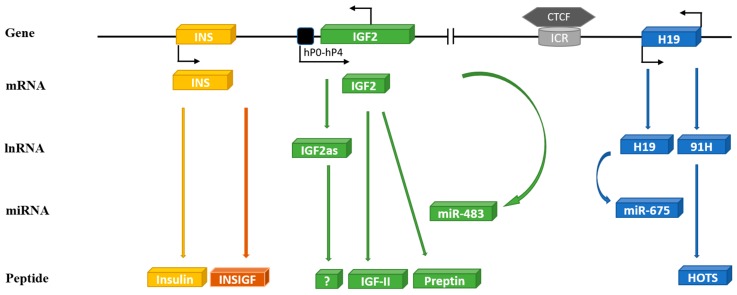
Schematic diagram of the human INS/IGF-II/H19 cluster genes on chromosome 11. The human IGF-II gene (consists of 10 exons) is transcribed into different mRNA transcripts originating from five unique promoters (P0-P4): IGF-II reverse transcribed yields IGF-II anti-sense (IGF-IIas) and micro RNA—miR-483. Human IGF-II mRNA translates to IGF-II peptide and preptin. The INS gene is located only 1.4 kb upstream from IGF-II consisting of three exons coding for insulin and INSIGF. The H19 gene is located 128kb downstream of IGF-II linked to it by an imprinting control region (ICR) to which a transcriptional regulator CCCTC-binding factor (CTCF) can bind and regulate imprinting of IGF-II/H19. The human H19 gene transcribes in a sense direction to yield long non-coding H19 and micro RNA—miR-675 or in antisense direction to 91H, which translates to H19 opposite tumor suppressor (HOTS).

**Figure 2 cells-08-01207-f002:**
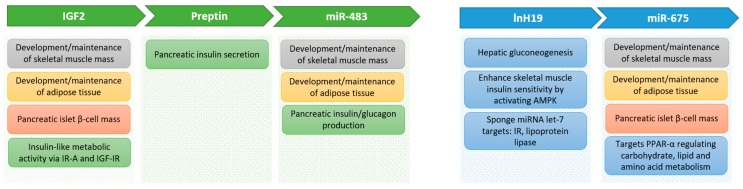
Metabolic effects of components of the Ins/Igf2/H19 cluster genes.

**Figure 3 cells-08-01207-f003:**
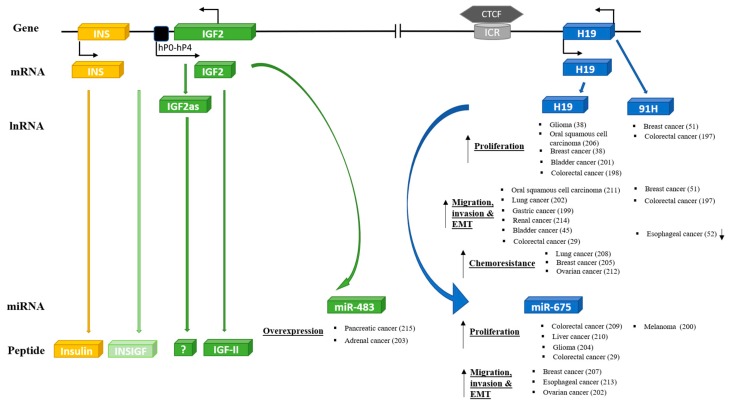
Oncogenic effects of other key components of INS/IGF-II/H19 cluster genes.

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
