# Peer review of "The Neglected Insulin: IGF-II, a Metabolic Regulator with Implications for Diabetes, Obesity, and Cancer"

_cells, 2019, doi:10.3390/cells8101207_

Round 1

Reviewer 1 Report

The review is well written, and provides a comprehensive overview of an often under-focused part of the field.

Small formatting edits are needed throughout to improve consistency, and at numerous points commas would help to break up long periods of text and improve flow for the reader. 

Examples:

Line 323: endoplasmic spelt wrong

line 555/6: inconsistent fonts

Line 242: subheading error

Line 703: is the word rest needed here?

Author Response

The review is well written, and provides a comprehensive overview of an often under-focused part of the field.

Small formatting edits are needed throughout to improve consistency, and at numerous points commas would help to break up long periods of text and improve flow for the reader. 

Thank you, I agree that the grammar was poor, this was written in a hurry before departing for a vacation and either I did not sufficiently proof-read, or I was too much in need of a vacation. I have now gone through the whole document and either split long sentences into two or used considerable more punctuation and also corrected other grammatical errors.

Examples:

Line 323: endoplasmic spelt wrong

line 555/6: inconsistent fonts

Line 242: subheading error

These errors have been corrected.

Line 703: is the word rest needed here?

The word ‘rests’ was from the original description in this old cited paper and I believe is appropriate in this context.

Reviewer 2 Report

Comments and Suggestions for Authors

The manuscript " The Neglected Insulin: IGF-II, a Metabolic Regulator with Implications for Diabetes, Obesity and Cancer", written by Jeff MP Holly, Kalina Biernacka and Claire M Perks.

The manuscript is a comprehensive and detailed review of the discovery and function of an important cluster of genes in the Ins/Igf2/H19; and is an appropriate addition to the literature summarizing the organization for IGF-II having an important role particularly in metabolic disorders and cancers. Although the paper is a review describing different roles of IGF-II in Diabetes, Obesity and Cancer, it also includes several problems as described below.

Comment 1:

A lot of evidence indicated potential roles for IGF-II in human physiology and consistently suggested IGF-II as a multifunctional metabolic regulator. The role of IGF-II in metabolic regulation is well described in this manuscript. There is no explanation regarding the interaction between IGF-II expression and glucose metabolism in cancer tissues. Since alternative splicing form of insulin receptor (IR-A) is the major signaling pathway of IGF-II, does it mean that IR-A is a good candidate for sensitizing cancer cells to the metabolic regulator?

In addition, Line 370-371 is a confusing sentence. Please modify or correct it.

Comment 2:

The authors discusse the function and regulation for IGF2-derived intronic miR-483-3p and miR-483-5p in Line 264-275. In Line 272, does the “miR-483p” mean miR-483-3p or miR-483-5p or miR-483?

Comment 3:

According to previous studies, IGF-II has an important role in the control of nutrient partitioning in the placenta and fetus in utero [131]. Besides, IGF-II may be the causal factor of diabetes. The insulin doesn’t transport across the placenta efficiently (PMID: 19467150). If the IGF-II/H19 related molecules could cross the placenta efficiently, such molecules may affect the growth pattern (central obesity and increased body weight) of the fetus in a mother with diabetes or gestational diabetes. Could IGF-II cross the placenta?

PMID: 19467150:

Hiden U, Glitzner E, Hartmann M, Desoye G. Insulin and the IGF system in the human placenta of normal and diabetic pregnancies. J Anat. 2009;215(1):60–68. doi:10.1111/j.1469-7580.2008.01035.x

Comment 4:

Authors have to carefully review the accuracy of website. For example: page 4, line 154, miRTarBase database is (mirtarbase.mbc.nctu.edu.tw) instead of (mirtarbase.mbc.nctu.edu).

Comment 5:

Page 6, Line 242: The “3.6.91. H” should be “3.6. 91H

In addition, some different grammar habits are noted throughout the manuscripts, especially the comma is absence in some sentences but present in other sentences. Consequently, those sentences are difficult to read. The followings are examples:

Page 3, Line 102-105: The original sentence lacks all commas. The sentence may better be “In contrast, abnormalities leading to gain of methylation at the ICR causes loss of imprinting (LOI) and over expression of IGF-II and down regulation of H19, which can result in Beckwith-Wiedeman Syndrome, an overgrowth disorder associated with neonatal hypoglycemia and an increased risk of childhood tumors [14,15].”

Page 5, Line 220-222: The original sentence lacks the comma. The sentence may better be “In this way, H19 inhibits E-cadherin expression in bladder cancer cells [45] and distinct subgroup of the Ras family member 3 (DIRAS3) in cardiomyocytes [46].”

Page 14, Line 633-635: The original sentence lacks the comma. The sentence may better be “In contrast, miR-675 was found to be consistently increased in a screen of miRNA that were altered during adipocyte differentiation from mesenchymal stem cells [157].”

Page 15, Line 683-685: The original sentence lacks the comma. The sentence may better be “In contrast, over-expression of IGF-II specifically in β-cells resulted in β-cell dedifferentiation and endoplasmic reticulum stress and increased the susceptibility of mice to diabetes; indicating that too much IGF-II expression in β-cells may predispose to the onset of diabetes [164].”

Page 18, Line 803-805: The original sentence lacks the commas. The sentence may better be “Similarly, the IGF-II molecular chaperone, GRP94, has been implicated in the proliferation, survival, invasion and metastasis of cancers as reviewed previously [214,215]; but again, the extent that these roles involve IGF-II is yet to be clarified.”

Please keep the grammar habits consistent in the manuscript.

Author Response

The manuscript " The Neglected Insulin: IGF-II, a Metabolic Regulator with Implications for Diabetes, Obesity and Cancer", written by Jeff MP Holly, Kalina Biernacka and Claire M Perks.

The manuscript is a comprehensive and detailed review of the discovery and function of an important cluster of genes in the Ins/Igf2/H19; and is an appropriate addition to the literature summarizing the organization for IGF-II having an important role particularly in metabolic disorders and cancers. Although the paper is a review describing different roles of IGF-II in Diabetes, Obesity and Cancer, it also includes several problems as described below.

Comment 1:

A lot of evidence indicated potential roles for IGF-II in human physiology and consistently suggested IGF-II as a multifunctional metabolic regulator. The role of IGF-II in metabolic regulation is well described in this manuscript. There is no explanation regarding the interaction between IGF-II expression and glucose metabolism in cancer tissues. Since alternative splicing form of insulin receptor (IR-A) is the major signaling pathway of IGF-II, does it mean that IR-A is a good candidate for sensitizing cancer cells to the metabolic regulator?

Thank you for this useful suggestion. We agree that it would strengthen the manuscript by re-emphasising the earlier section on IR-A again in the section on cancer and how it relates to metabolism in cancer tissues. We have added several sentences to this effect on lines 888-895.

In addition, Line 370-371 is a confusing sentence. Please modify or correct it.

We have modified this sentence to make it more clear.

Comment 2:

The authors discusse the function and regulation for IGF2-derived intronic miR-483-3p and miR-483-5p in Line 264-275. In Line 272, does the “miR-483p” mean miR-483-3p or miR-483-5p or miR-483?

We believe that the correct nomenclature is used here as in the cited work.

Comment 3:

According to previous studies, IGF-II has an important role in the control of nutrient partitioning in the placenta and fetus in utero [131]. Besides, IGF-II may be the causal factor of diabetes. The insulin doesn’t transport across the placenta efficiently (PMID: 19467150). If the IGF-II/H19 related molecules could cross the placenta efficiently, such molecules may affect the growth pattern (central obesity and increased body weight) of the fetus in a mother with diabetes or gestational diabetes. Could IGF-II cross the placenta?

PMID: 19467150:

Hiden U, Glitzner E, Hartmann M, Desoye G. Insulin and the IGF system in the human placenta of normal and diabetic pregnancies. J Anat. 2009;215(1):60–68. doi:10.1111/j.1469-7580.2008.01035.x

Thank you for this useful suggestion, we have added a new paragraph regarding the potential consequences of IGF-II actions on the placenta in relation to metabolic disorders on lines 612 to 621 with new references, including the one suggested.

Comment 4:

Authors have to carefully review the accuracy of website. For example: page 4, line 154, miRTarBase database is (mirtarbase.mbc.nctu.edu.tw) instead of (mirtarbase.mbc.nctu.edu).

Thank you, we have corrected this.

Comment 5:

Page 6, Line 242: The “3.6.91. H” should be “3.6. 91H

We have corrected this.

In addition, some different grammar habits are noted throughout the manuscripts, especially the comma is absence in some sentences but present in other sentences. Consequently, those sentences are difficult to read. The followings are examples:

Page 3, Line 102-105: The original sentence lacks all commas. The sentence may better be “In contrast, abnormalities leading to gain of methylation at the ICR causes loss of imprinting (LOI) and over expression of IGF-II and down regulation of H19, which can result in Beckwith-Wiedeman Syndrome, an overgrowth disorder associated with neonatal hypoglycemia and an increased risk of childhood tumors [14,15].”

Page 5, Line 220-222: The original sentence lacks the comma. The sentence may better be “In this way, H19 inhibits E-cadherin expression in bladder cancer cells [45] and distinct subgroup of the Ras family member 3 (DIRAS3) in cardiomyocytes [46].”

Page 14, Line 633-635: The original sentence lacks the comma. The sentence may better be “In contrast, miR-675 was found to be consistently increased in a screen of miRNA that were altered during adipocyte differentiation from mesenchymal stem cells [157].”

Page 15, Line 683-685: The original sentence lacks the comma. The sentence may better be “In contrast, over-expression of IGF-II specifically in β-cells resulted in β-cell dedifferentiation and endoplasmic reticulum stress and increased the susceptibility of mice to diabetes; indicating that too much IGF-II expression in β-cells may predispose to the onset of diabetes [164].”

Page 18, Line 803-805: The original sentence lacks the commas. The sentence may better be “Similarly, the IGF-II molecular chaperone, GRP94, has been implicated in the proliferation, survival, invasion and metastasis of cancers as reviewed previously [214,215]; but again, the extent that these roles involve IGF-II is yet to be clarified.”

Please keep the grammar habits consistent in the manuscript.

Thank you, I agree that the grammar was poor, this was written in a hurry before departing for a vacation and either I did not sufficiently proof-read, or I was too much in need of a vacation. We have amended the sentences as suggested and have now gone through the whole document and either split long sentences into two or used considerable more punctuation and also corrected other grammatical errors.